



# Real-time pollen monitoring using digital holography

Eric Sauvageat[1,2*], Yanick Zeder[3,4*], Kevin Auderset[5], Bertrand Calpini[1], Bernard Clot[1], Benoît Crouzy[1], Thomas Konzelmann[1], Gian Lieberherr[1], Fiona Tummon[1], and Konstantina Vasilatou[5]

[*]Yanick Zeder and Eric Sauvageat contributed equally to this work.
[1]Federal Office of Meteorology and Climatology MeteoSwiss, Payerne, Switzerland
[2]University of Bern, Bern, Switzerland
[3]Lucerne University of Applied Sciences and Arts, Lucerne, Switzerland
[4]Swisens AG, Horw, Switzerland
[5]Swiss Federal Institute of Metrology METAS, Bern-Wabern, Switzerland

**Correspondence:** Eric Sauvageat (eric.sauvageat@iap.unibe.ch)

**Abstract.** We present the first validation of the Swisens Poleno, currently the only operational automatic pollen monitoring system based on digital holography. The device provides in-flight images of all coarse aerosols and here we develop a two-step classification algorithm that uses these images to identify a range of pollen taxa. Deterministic criteria based on the shape of the particle are applied to initially distinguish between intact pollen grains and other coarse particulate matter. This first level

of discrimination identifies pollen with an accuracy of 96 %. Thereafter, individual pollen taxa are recognised using supervised learning techniques. The algorithm is trained using data obtained by inserting known pollen types into the device and out of eight pollen taxa six can be identified with an accuracy of above 90 %. In addition to the ability to correctly identify aerosols, an automatic pollen monitoring system needs to be able to correctly determine particle concentrations. To further verify the device, controlled chamber experiments using polystyrene latex beads were performed. This provided reference aerosols with

traceable particle size and number concentrations to ensure particle size and sampling volume were correctly characterised.

## 1 Introduction

The incidence of pollinosis and related diseases has increased considerably over the past decades, sparking growing research interest into aeroallergens and pollen monitoring. Among aeroallergens, pollen is the most important impacting approximately

20 % of the population in Switzerland and other high income countries (D'Amato et al., 2007; Wüthrich et al., 1995; Ring et al., 2001). Most often, sensitized patients exposed to allergenic pollen experience symptoms of allergic rhinitis or hay fever, but exposure to pollen has also been shown to exacerbate the development of more severe diseases like asthma, all of which have significant effects on public health and the economy (Greiner et al., 2012; Gamble et al., 2008).

Beyond the issue of public health, the airborne transport of pollen plays a key role in ecosystem dynamics, with important

implications for agriculture, forestry, and the geographic dispersion of plants (Garzia-Mozo, 2011; Oteros et al., 2014). The



relevance of pollen and other bioaerosols for atmospheric chemistry and physics has also been increasingly acknowledged since they represent a significant fraction of atmospheric particulate matter and have been shown to influence cloud formation and precipitation (Jaenicke, 2005; Pöschl, 2005; Möhler et al., 2007; Deguillaume et al., 2008; Pope, 2010). Furthermore, in the context of climate change, pollen concentrations undergo fluctuations in terms of taxa, abundance, and seasonal trends. Pollen

monitoring thus provides valuable information about the evolution of the local biosphere and its response to anthropogenic forcings such as pollutant emission or intensive urbanization. While still uncertain, some evidence shows that the combination of a globally warming climate and the perpetuation of contemporary human lifestyle is very likely to increase the prevalence, intensity, and related costs of pollen-related allergenic diseases in the coming decades (Ring et al., 2001; D'Amato et al., 2001, 2016; Beggs, 2016).

Airborne pollen has been monitored since the mid-twentieth century in Switzerland and elsewhere in Europe (Clot, 2003; Spieksma, 1990), most commonly with Hirst-type samplers (Hirst, 1952). These instruments continuously collect airborne particles on a rotating cylinder tape which is then collected and pollen particles manually identified and counted using optical microscopy, typically on a weekly basis. Because this is such a time- and labour-intensive method, the spatial and temporal resolution of the measurements is severely limited. Another drawback of this type of sampler is the inevitable delay between

the observations and their analysis (up to 9 days), which has important implications in terms of pollen forecasts. In particular, the availability of real-time data with high temporal resolution is a key step (Sofiev, 2019) in the development of accurate forecasting models for atmospheric pollen transport (Pasken and Pietrowicz, 2005; Schueler et al., 2006; Sofiev et al., 2006; Vogel et al., 2008; Zink et al., 2013; Sofiev, 2017). More accurate predictions would represent a tremendous asset for both the scheduling of patients' activities and the planning of their medical treatment.

To respond to the need for real-time pollen information, numerous partly- or fully-automated monitoring systems have been developed and investigated over the past decade, with some recently having reached an operational level. Among the existing devices on the market, two main categories of instruments can be identified in terms of the different techniques utilised, either microscope-based or in-situ measurements (Kawashima et al., 2007; Perring et al., 2015; Oteros et al., 2015; Crouzy et al., 2016; Šauliene et al., 2019). The former aim to automatise the microscopic analysis process, while the latter make use of air-

flow cytometry measurements, avoiding the collection step and performing real-time particle-by-particle identification. In the category of air-flow cytometers, most existing devices rely on fluorescence and elastic light-scattering measurements combined with machine-learning algorithms to identify and quantify airborne pollen concentrations. Some of these systems have already shown promising results and are currently tested in different European countries (Crouzy et al., 2016; Šauliene et al., 2019).

In this paper we evaluated a new automated pollen monitoring system based on air-flow cytometry, the Swisens Poleno.

This device captures holographic images of each airborne particle in addition to measurements of optical properties such as fluorescence intensity, lifetime, and elastic light scattering. Here, we focus on the use of digital holography for online pollen monitoring since this technique allows a certain degree of visual identification of pollen taxa. We use a combination of classical image analysis and a neural network algorithm to assess the instrument's performance in terms of pollen identification compared to manually-classified calibration sets. Aerosol sampling, particle sizing and counting performance are evaluated

using a reference particle counter at the Swiss Federal Institute of Metrology, METAS (Horender et al., 2019).





In the following section, the Swisens Poleno air flow cytometer is presented as well as the methodology used for the data analysis. Thereafter, the performance achieved in pollen identification and counting using holographic images from the device is shown. Although the focus of the present paper lies on the use of digital holography to identify pollen grains, a validation of the output of the fluorescence using standard particles is also performed. Finally, the significance of the results for pollen

monitoring are discussed and an overview of the future perspectives for this new technology is provided.

## 2 Materials and Methods

### 2.1 Swisens Poleno

We used the first unit of the commercially-available Poleno device developed by Swisens AG (Switzerland). The device provides in-flight measurement of particle shape, size, and fluorescence using various light sources and detectors. The schematic

structure of the device is presented in Figure 1. Laser light scattering triggers the measurement, together with providing a first estimation of particle size, velocity, and alignment by combining the information of two trigger lasers. Following the trigger, two focused images at 90° from each other are reconstructed using digital holography as in Berg and Videen (2011), and UV-induced fluorescence produces information regarding the particle composition. UV-induced fluorescence lifetime and spectra are measured at three different excitation wavelengths (280, 365, and 405 nm) using five measurement emission windows

between 320 and 720 nm. Finally, a measurement of the time-resolved optical polarization characteristics of the particle is acquired before it exits the device.





**Figure 1.** Measurement principle of the Swisens Poleno (courtesy Swisens AG).

Since atmospheric pollen concentrations are typically low compared to other aerosols (on the order of 10-10 000 pollen grains per cubic metre), a high sampling rate is necessary to sample pollen effectively. This is all the more relevant since the threshold for allergic response is typically even lower, varying depending on the taxa, from just a few grains to a few tens of grains per cubic metre. In the Swisens Poleno, this level of sampling is achieved using a concentrator based on the principle of a virtual impactor that enables an effective flow rate of 40 litres per minute. An hourly time resolution providing





concentrations relevant for pollen exposure thresholds can thus easily be achieved with this sampling rate. The drawback, however, is that the saturation level occurs at coarse particle (>10 µm) concentrations above 30 000 particles per cubic metre. Note that size-dependent particle loss occurs in the concentrator: Corrected concentration factors were determined from the

controlled chamber experiments presented at the end of the manuscript. A Sigma2 inlet was chosen to protect the device from precipitation as its sampling, in particular the role of wind speed and particle size, has been documented in detail (Verein Deutscher Ingenieure, 2013).

## 2.2 Calibration dataset

A large (> 750 particles per pollen taxon) calibration dataset was collected for eight different pollen taxa using online measure-

ments from the Swisens Poleno device. The taxa were chosen to present a good range of particle size - from small nettle pollen grains through to large pine pollen grains - and morphology. Note that the list includes taxa relevant for pollen allergies such as two different Betulaceae, *Dactylis glomerata* as a proxy for grasses, and ash. These samples were used to train a machine learning algorithm applied to identify the different pollen taxa. Only one particle type was calibrated at a time, allowing the data points to be labelled directly, although dirt, debris and agglomerates needed to be eliminated from the dataset manually

through visual inspection of the holographic images. To generate a large number of events without saturating the detector, pollen samples were continuously aerosolised using sound waves in a closed chamber around the detector inlet. Figure 2 shows examples of the reconstructed images generated for the calibration dataset. The pollen identification presented here is based just on these reconstructed images since they are expected to contain enough relevant morphological information to permit sufficient identification of the taxa of interest. Fluorescence and lifetime measurements are expected to be pertinent for extend-

ing the scope of the device to characterise other bioaerosols (e.g. spores) and pollutants. The dataset obtained includes 12234 pollen grains (two images per grain) and is summarised in Table 1; 80 % of this dataset was used for algorithm calibration and 20 % for validation purposes. The images are grey scale and have a resolution of 200x200 pixels. Each pixel represents a 0.56 µm by 0.56 µm physical domain.





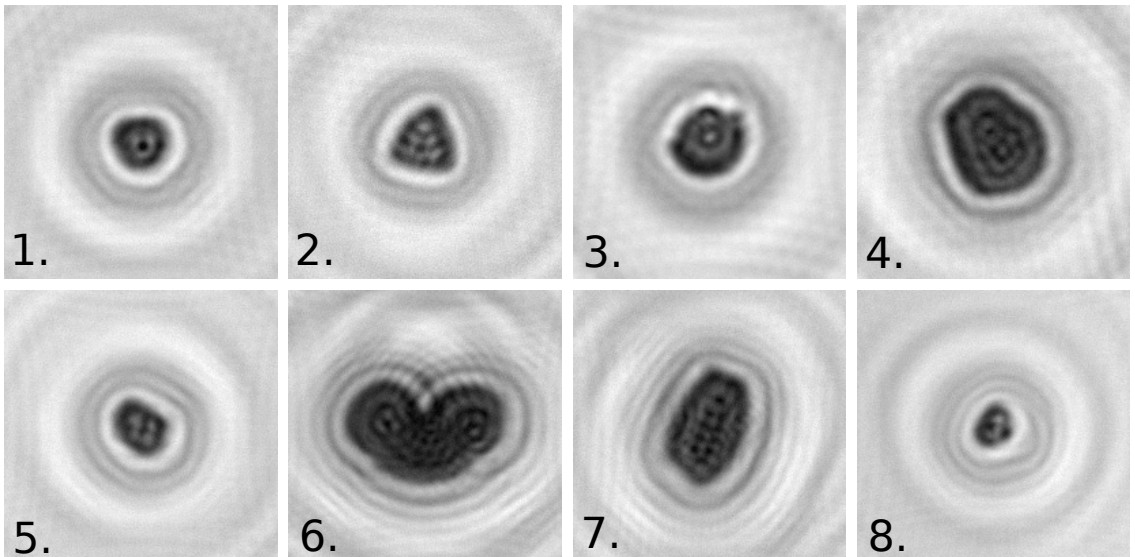

**Figure 2.** Reconstructed holographic images from the Swisens Poleno for different pollen taxa: 1. *Ambrosia artemisiifolia*, 2. *Corylus avellana*, 3. *Dactylis glomerata*, 4. *Fagus sylvatica*, 5. *Fraxinus excelsior*, 6. *Pinus sylvestris*, 7. *Quercus robur*, 8. *Urtica dioica*.

**Table 1.** List of pollen taxa used to train the classification algorithm, including the number of events used for training and validation of the machine learning algorithm for each taxa. Note that all pollen were in a dry state.

| Common Name | Taxa (Latin) | Supplier | # Training events | # Validation events |
|---|---|---|---|---|
| Ragweed | *Ambrosia artemisiifolia* | Bonapol | 1063 | 266 |
| Hazel | *Corylus avellana* | Bonapol | 1156 | 289 |
| Grasses | *Dactylis glomerata* | Bonapol | 602 | 151 |
| Beech | *Fagus sylvatica* | Allergon | 859 | 215 |
| Ash | *Fraxinus excelsior* | Allergon | 826 | 207 |
| Pine | *Pinus sylvestris* | Bonapol | 3601 | 901 |
| Oak | *Quercus robur* | Bonapol | 775 | 194 |
| Nettle | *Urtica dioica* | Bonapol | 903 | 226 |

## 2.3 Shape analysis for pollen detection

A large range of coarse aerosol particles were seen in the events recorded by the Swisens Poleno. To provide a clean dataset to the pollen classification algorithm, the pollen particles needed to be discriminated from all others. In principle this can be


done by applying thresholds to the confidence estimates provided by deep-learning pollen-classification algorithms, however, this simple method did not yield the required level of accuracy. An additional step was therefore implemented in the algorithm (thus becoming a two-step classifier), using shape analysis to discriminate between pollen and non-pollen particles prior to applying the full pollen classification.

In general, unbroken biological particles tend to have a smooth, convex shape while dust, debris, or other non-biological particles have rougher, more chaotic shapes (see, for example, Figure 3). Two deterministic image analysis routines were developed and evaluated to select the best available method for distinguishing pollen from other detected particles. Both use the contour of each particle, which is extracted from the reconstructed holographic images in three steps: (1) pixels are separated into two classes using the Otsu binarisation algorithm (Otsu, 1979), which is based on a dynamic intensity threshold; (2) the largest cluster corresponding to the particle of interest is then identified; and (3), a convolution operation extracts the contour of the particle.

The first routine for biological particle identification uses the OpenCV2 library (Bradski, 2000) to fit (in a least-squares sense) an ellipse to each contour (Fitzgibbon et al., 1999). As a feature for biological particle identification the fraction $f_c$ of the contour located further than a certain distance from the fitted ellipse is considered (red pixels in Figure 3). For pollen grains, this value is typically low, while for more fragmented particles this fraction can reach up to 60 % of the contour (0 % and 46 % respectively for the examples shown in Figure 3).

The second method is based on the fractal dimension, which characterises the state of self-similarity or roughness, and is also estimated from the particle contour. Such analysis of natural objects was first introduced by Mandelbrot (1983), and is now widely used in a variety of applications, such as plant leaf or sediment identification (Backes et al., 2009; Orford and Whalley, 1983). We make use of the so-called box-counting algorithm to estimate the fractal dimension of the holographic images because of its computational simplicity (Theiler, 1990). This method consists of splitting each image into grid boxes and counting the number of boxes $N(s)$ that contain a fragment of the particle's perimeter, where $s$ is the box size. By repeating the procedure for decreasing values of $s$, the fractal dimension is then estimated by computing the slope of $log(N(s))$ against $log(\frac{1}{s})$ (see lower panel, Figure 3).

The performance of these two methods (ellipse fitting and fractal dimension) is compared using the reference dataset which contains images from all calibrated pollen taxa (1640 particles) and manually-selected coarse aerosols (1554 non-pollen particles), measured during late spring and summer 2018 in Payerne, Switzerland. Smaller particles were filtered out prior to this comparison to keep only particles roughly in the pollen size range (10 to 200 μm). Note that only a reduced subset of pollen calibrations was used to keep a balanced dataset with respect to the non-pollen category.

Using manually-labelled data to verify the output of the classification algorithm, we performed a grid optimization to find the set of parameters that best discriminates between pollen and other airborne particles, that is, a filter that simultaneously provides sufficient recall (ensuring that most pollen grains are classified as pollen) and precision (ensuring that only pollen grains are classified as pollen).




**Figure 3.** Illustration of the image analysis routines applied to a pollen grain (*Quercus robur*; left) and coarse particulate matter (right).





### 2.4 Pollen classification using deep learning

Developments in computer hardware have made it possible to perform efficient training of complex artificial, or 'deep', neural networks; their use in image recognition problems is the iconic application of deep learning. Mimicking the visual cortex, a series of so-called *convolutional layers* identify relevant patterns and concentrate the information diluted over a large image. Extracted features are then used as input for fully-connected layers of artificial neurons, which combine the features to determine associated labels for each image. This technique is part of the family of supervised-learning algorithms; networks need to be trained using images for which the label is known. We used the open source software library Keras (Chollet, 2015) with TensorFlow (Abadi et al., 2015) as computational back-end to implement the pollen identification algorithm.

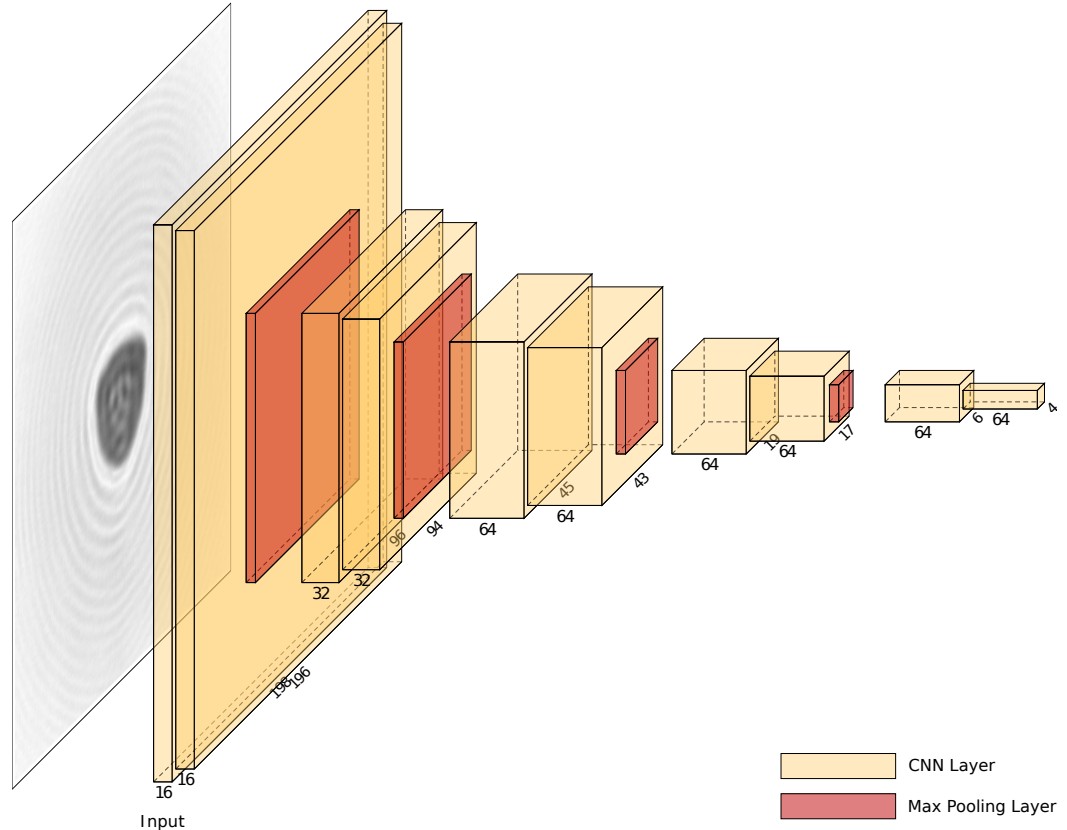

**Figure 4.** Vision model based on the VGG16 architecture (Simonyan and Zisserman, 2014) as used here for pollen classification.

The model used to classify pollen grains is based on the VGG16 architecture (Simonyan and Zisserman, 2014), which has successfully been applied to a wide range of different image classification problems (Russakovsky et al., 2015). The basic model is shown in Figure 4, with the vision model being applied separately to the two orthogonal images and the output then being processed with two fully-connected layers. This ensures that the model is able to use the information from both



images. For the final layer, softmax activation is used to map the network output to a probability distribution (Chollet, 2015). The predicted pollen label is determined by taking the most probable class. Note that probability information is also useful since the plausibility of the final classifications can be easily verified (Crouzy et al., 2016; Šauliene et al., 2019). Furthermore,

although not carried out here, a threshold could be applied to the classification when performing operational measurements to retain only the pollen grains classified above a sufficient confidence level.

## 3 Results and Discussion

### 3.1 Pollen identification

The discrimination between pollen and other coarse aerosols is evaluated in Figure 5 in the form of a normalised confusion

matrix for each of the two image analysis algorithms. Each line in Figure 5 is normalised to one and the values along the diagonal provide the recall for each category.

  The convexity hypothesis for unbroken biological particles seems to hold particularly well for pollen grains. Indeed, in nearly all cases one of the two images has an almost perfect elliptical fit for pollen particles, which translates into only a very small fraction of contour pixels that strongly deviates from the optimal ellipse, i.e. $f_c \approx 0$. Other particles most often present

values of $f_c \geq 20$ % for both images. Good precision can be obtained by using a low $f_c$ threshold, albeit at the expense of recall. The best results (achieving both good precision and recall) were obtained by imposing different thresholds on the two images: at least one of the two images needs to satisfy a hard condition on the value of $f_c$, while the other should not present excessive deviation from its fitted ellipse. When using optimal values of the two $f_c$ thresholds, an overall accuracy of 96 % was achieved for the discrimination of pollen from other particles.

Visually, the contours of pollen grains clearly exhibit smoother shapes than non-pollen particles. However, it can be noted from Figure 5, that the fractal dimension method did not function as well as the ellipse fitting one. While an overall accuracy of 77 % is reached, the number of non-pollen particles classified as pollen (false positives) is still too high to ensure a satisfactory classification in the second step of the deep-learning algorithm. Note that the ellipse fitting algorithm alone performs better than any combination of the two methods (not shown here).



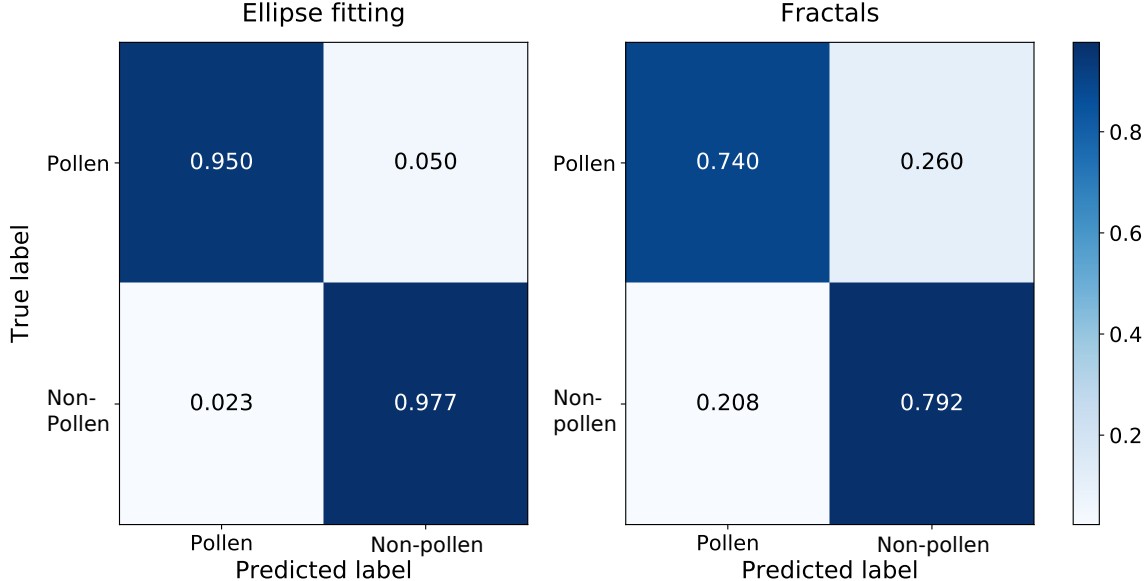

**Figure 5.** Normalised confusion matrix summarizing the performance of the ellipse-based classifier (left) and fractal-based classifier (right). 'Pollen' refers to a mix of different pollen taxa while 'Non-pollen' encompasses all other coarse aerosols.

Estimating the fractal dimension of an object from a holographic image is sensitive to the image resolution, which is thought to have a significant influence on the determination of the fractal dimension (see Baveye et al. (1998)). Indeed, more detailed images tend to improve the estimation of an object's fractal dimension as more details of the contour become apparent. Furthermore, the binarisation process (i.e. reducing the grey scale holographic image to black and white) may also affect the box-counting calculation. Should higher resolution images be available in future versions of the Poleno, the fractal dimension

method may be worth implementing. At this point, given the better accuracy of the ellipse-fitting technique, we utilise this method in the final algorithm.

### 3.2   Pollen classification

Once a particle has been identified as a pollen grain it needs to be classified into the right taxa. Using the convolutional neural network (CNN) described in Section 2.4, each airborne particle is assigned a taxa with a corresponding confidence level of

prediction. Results from the classification model are presented as a normalised confusion matrix in Figure 6. The sum of each line is normalised to one, and the diagonal values indicate the recall for the different pollen taxa.

   Overall the classification algorithm performs very well, with 6 of the 8 pollen taxa being classified with an accuracy of over 90 %. The exceptions are *Corylus*, which is confused in 10 % of cases with *Fraxinus*, ad *Dactylis*, which is confused 22 % of the time with *Corylus*. Note that in this regard the problem presented to the algorithm is somewhat artificial: *Corylus*

and grass pollen are not likely to be simultaneously present in the atmosphere in concentrations relevant for pollen allergies.





Nevertheless, a larger mix of pollen taxa is likely to be observed in reality, highlighting the need for further developments to the classification algorithm using a larger number of species and including fresh pollen.

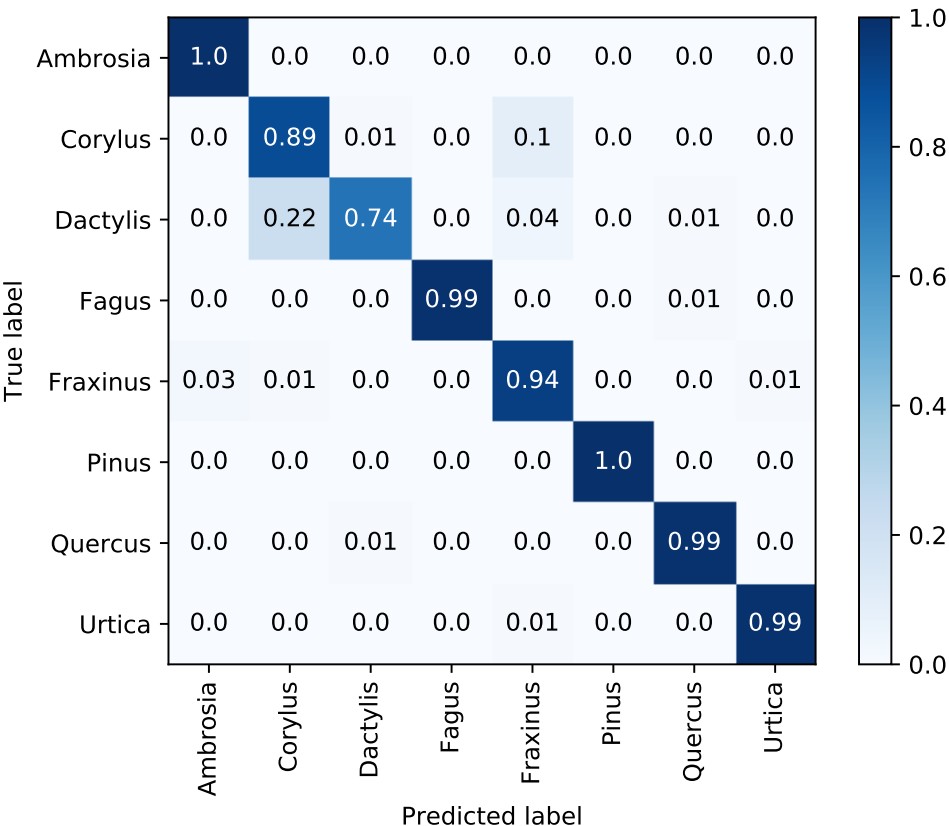

**Figure 6.** Normalised confusion matrix for the pollen taxa identification, the second step of the classification algorithm.

To better understand the functioning of the neural network, Figure 7 presents activation heatmaps (Kotikalapudi and contributors, 2017) of pollen particles. These show which parts of the image the network focuses on to make the taxa prediction; in our case, strongly on the particle shape. This is apparent in the heatmaps (Figure 7), as the highest activation regions follow the outline of the pollen grains. This may appear as an obvious result but confirms the validity of the CNN step of the classification algorithm and indicates that the predictions are based on a physical feature of the particle and not on some other information embedded in the images. This verification is essential, as differences in light intensity, or the presence of dust on lenses, could lead to discrimination between calibrations not based on pollen morphology but on artifacts.






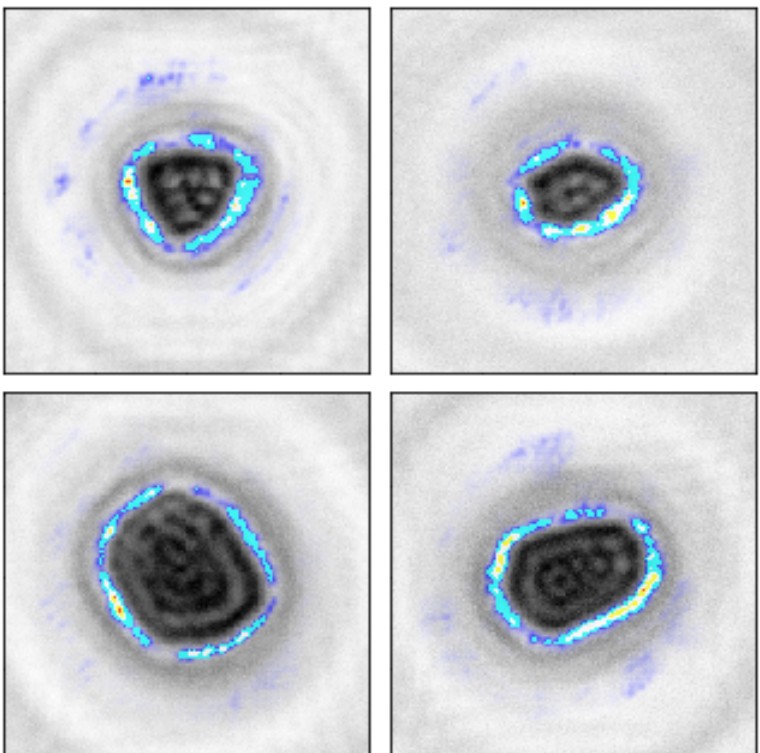

**Figure 7.** Visualisation of the areas on which the convolutional neural network for pollen classification focuses.

Although there are some limitations to the use of dry pollen for model training purposes, the performance obtained suggests that holography alone is sufficient to distinguish between different pollen taxon. Combined with the results of the previous section on pollen identification, we propose a two-step approach for operational pollen monitoring using digital holography, first applying classical image analysis to identify pollen, and subsequently using deep learning to classify these particles into individual pollen taxa. As mentioned in Section 2.4, the identification algorithm provides a measure of confidence in addition to the predicted label. Note that raw results are presented in the confusion matrix (Figure 6); in an operational setup confidence thresholds could be used to increase precision further. Due to the large sampling of such an automatic system, a certain loss of particles from introducing confidence thresholds can be accepted without losing statistical significance of the sampling.

### 3.3 Reference particle counts and fluorescence observations

The focus of this study was to assess the performance of the Swisens Poleno in terms of pollen identification. While this is key, it is equally as important to accurately quantify airborne pollen concentrations. At present, this remains a difficult task since no method, standardised or other, exists to aerosolise a known quantity of a known pollen taxa. Pollen grains are both considerably larger than other, non-biological aerosol particles and relatively fragile, so producing homogenised airborne concentrations is currently not possible with conventional techniques.





To assess the accuracy of the particle concentrations obtained with the Swisens Poleno, a measurement campaign was carried out at the Swiss Federal Institute of Metrology (METAS). The custom-made facility at METAS has been described in detail in Horender et al. (2019). The aim was to compare the Poleno device with reference particle concentrations and fluorescence observations in a controlled calibration chamber using Polystyrene Latex (PSL) spheres. Different sizes, ranging from 0.5-20 µm, were tested along with 3 types of fluorescent PSL (Blue 2.07 µm, Plum Purple 2.07 µm, and Red 2.07 µm) to provide a first insight into the quality of the fluorescence measurements. For each size, the concentrations measured by the Poleno were compared to the reference concentrations for approximately 20 minutes. The fluorescent PSL used here have been fully characterised by the Max Planck Institute for Chemistry (MPIC) (Könemann et al., 2018) for a large range of excitation wavelengths. Those corresponding to the Poleno excitation wavelengths have been reproduced in Figure 8 and serve as a reference for the fluorescence measurements. Since fluorescence intensity is measured in arbitrary units a.u., the fluorescence measured by the Poleno (filled dots) is scaled to the MPIC reference values (solid lines) using the maximum for each of the 5 emission windows located between 335 and 700 nm.



**Figure 8.** Top: Concentrations (5 and 10 µm PSL) scaled to the METAS reference measurements. Bottom: Comparison of fluorescence measurements. Solid lines are the reference fluorescence intensities measured by the Max Planck Institute of Chemistry presented in Könemann et al. (2018). Median measurements from the Poleno are shown with error bars (interquartile range). Each excitation wavelength is scaled individually (see text for details).





The results presented here are encouraging, both in terms of particle concentration and fluorescence measurements. The Poleno seems to follow the fluctuations in terms of particle concentration very well, with Pearson correlation values of 0.905 and 0.916 for the 5 μm and 10 μm sizes, respectively (see Figure 8). Similar results are observed for the other particle sizes tested (not shown), indicating that the Poleno measures the size of the certified PSL particles correctly. It is important to note, however, that the Poleno values have been scaled to the METAS values since the particle concentrator is size selective, with larger particles being better sampled. Once a size-scaling curve has been established it can be effectively applied to all future measurements, which is a significant improvement compared to the current practice of deriving scaling factors for automatic pollen monitors from Hirst-type measurements (Crouzy et al., 2016; Šauliene et al., 2019). The systematic analysis of the efficiency of the concentrator goes beyond the scope of this paper, but will be described in future work. The reproducibility of the scaling factors obtained was verified by repeating the experiments with the 2 μm particles three times.

Despite the fact that the Poleno does not measure a continuous fluorescence emission spectrum, Figure 8 confirms that it already provides an insight into the shape of the spectra for the different excitation wavelengths. The Poleno fluorescence signals agree well with the off-line reference measurements performed at MPIC (Könemann et al., 2018) for all 5 emission windows and combined with the holographic images, potentially provide the opportunity to extend the number of particle types that can be recognised (e.g. further pollen taxa, spores, or pollutants).

## 4  Towards operational pollen monitoring

The focus of this study was to assess the performance of the Swisens Poleno, the first operational automatic pollen monitoring system based on digital holography. The potential of using these in-flight images to classify pollen particles in real-time was shown for 8 pollen taxa using a two-step classification algorithm. The first step distinguishes intact pollen grains from other coarse aerosol particles using a deterministic ellipse-fitting method, providing a 96 % discrimination accuracy for pollen. Thereafter, individual pollen taxa are recognized using supervised learning techniques. The algorithm is trained using data obtained by inserting known pollen types into the device and 6 out of 8 pollen taxa can be identified with an accuracy of above 90 %.

The ability of the device to accurately count particles was tested against reference measurements in controlled chamber experiments using polystyrene latex spheres. This is a key aspect for any monitoring device that is to be used operationally and to date has not been accurately assessed. These tests, together with validation of the fluorescence measurements carried out in the same chamber, provide very promising results.

The holographic images open the possibility for a human expert to perform online training and improve the model through a feedback loop. This effectively means that falsely-classified pollen are identified manually and put into the correct class, for the model to use in the next training phase. The same principle could potentially be applied when the device is deployed in a new region with different pollen taxa by creating new pollen classes. Since the Swisens Poleno measures one cubic metre of air every 25 minutes, such new datasets can be created relatively quickly.



Finally, while not included in this study, the use of the fluorescence observations may provide the potential to identify particles other than pollen, for example, spores or other pollutants. This could lead to synergies with air pollution monitoring

networks and be of significant benefit to other sectors, such as agriculture and forestry, where real-time information concerning the distribution of spores could lead to better crop management practices. Future work in this direction is being continued, as is the development of the machine learning algorithm to identify further pollen taxa.

*Code and data availability.* All data and algorithms presented in the paper are experimental and subject to further development. They are available for research purposes on request to the authors of the paper. Work is in progress to harmonize the algorithms and make them public

together with the data via open software and data repositories.

*Author contributions.* BCl, BCr, ES and YZ designed the study. BCr, KV, KA, and FT carried out the METAS campaign. ES and YZ analysed all available data. ES, YZ, BCr, and FT prepared the paper with contributions from all other authors.

*Competing interests.* The authors declare that they have no conflict of interest.



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
