# Peer review of "Real-time pollen monitoring using digital holography"

_Atmospheric Measurement Techniques, 2019_

## Referee Comment (RC1) · Anonymous Referee #1 · 18 Dec 2019

The work presented in this article, is very interesting and the use of "digital holography" is a great novelty, but I will make some considerations, since it is a scientific work and not only the description of a new real-time sampling system: Introduction lines. 40-41 "To respond to the need for real-time pollen information, numerous partly- or fully-automated monitoring systems have been developed and investigated over the past decade, with some recently having reached an operational level" The authors should extend the introduction, describing succinctly the monitoring systems have been developed in the past decade and what physical principles or algorithms have been based on, only mention some, with the comment that it is the only thing that has been done in Europe. Other bioaerosol sampling has been carried out with different automated methods (e.g. Kawashima et al. 2016, Savage et al. 2017) and specifically in Europe

with the Wibs system (e.g. O'Connor et al. 2015, Calvo et al. 2018).

Results and discussion

3.2. Pollen classification

Lines 184. . ... "Note that in this regard the problem presented to the algorithm is somewhat artificial: Corylus and grass pollen are not likely to be simultaneously present in the atmosphere in concentrations relevant for pollen allergies". This assertion is not entirely true, there are European regions where hazelnut and grass blooms coincide. On the other hand, in this work the Betula pollen type is not mentioned, with morphological characteristics and of dimensions practically equal to Corylus and that if it is present in the atmosphere at the same time as grasses, in very high concentrations and sufficient to cause allergic responses. It is not likely that Swispoleno can differentiate these types through digital holography. But it would be very important to point out this fact, since Betula is the pollen type that causes the most allergy in Central Europe, along with the grasses

3.3 Reference particle counts and fluorescence observations

Line 205-208. "At present, this remains a difficult task since no method, standardised or other, exists to aerosolise a known quantity of a known pollen taxa". "Pollen grains are both considerably larger than other, non-biological aerosol particles and relatively fragile, so producing homogenised airborne concentrations is currently not possible with conventional techniques" In relation to these two paragraphs and others of the same section 3.3, the authors should review in greater depth certain bibliography related to the methods of differentiation of biological particles and quantification (among other pollen and spores) by fluorescence (e.g. Toprak and Schnaiter 2013, Hernández et al. 2016, Savage et al. 2017)

4 Towards operational pollen monitoring

"The focus of this study was to assess the performance of the Swisens Poleno, the first

operational automatic pollen monitoring system based on digital holography". Certainly digital holography is the first time it has been used in bioparticle monitoring systems (specifically pollen) in real time, but the authors need to make a deep and detailed discussion, highlighting the advantages, similarities or great differences with other types of bioaerosol monitoring systems in real time. As I point out with some bibliographical references, certain pioneering works in this field are not mentioned

Finally, making few modifications to the introduction and discussion, the article can be published without problems, because I consider that the work constitutes a good scientific contribution, and a good basis to continue working in this field, as the authors propose

---

## Referee Comment (RC2) · Anonymous Referee #2 · 12 Feb 2020

This article presents the development and testing of a commercial instrument to characterize pollen aerosol particles with fluorescence and digital holography. The holography aspect is based on previous work by Berg & Videen in 2011 where images of similar free-flowing pollen particles are obtained revealing the particle size and shape. The authors integrates this powerful technique with fluorescence to collect material information from the particles at a later stage of trajectory through the instrument. This is important work as it demonstrates that the earlier proof-of-principle research can be effectively implemented as an instrument. The paper's presentation is of high quality and its impact will be substantial to the field of aerosol science.

---

## Author Comment (AC1) · 13 Feb 2020

Many thanks to reviewer 2 for the positive feedback, we really appreciate it!
* * *

---

## Author Response (AR1)

**Response to reviewers**:

We would like to thank reviewer 1 for the positive comments and feedback that have helped to improve the manuscript. Below please feed the response to each reviewers comment (in italics) as well as the sections of manuscript that were adapted (in blue).

*The work presented in this article, is very interesting and the use of "digital holography" is a great novelty, but I will make some considerations, since it is a scientific work and not only the description of a new real-time sampling system: Introduction lines. 40-41 "To respond to the need for real-time pollen information, numerous partly- or fully automated monitoring systems have been developed and investigated over the past decade, with some recently having reached an operational level". The authors should extend the introduction, describing succinctly the monitoring systems have been developed in the past decade and what physical principles or algorithms have been based on, only mention some, with the comment that it is the only thing that has been done in Europe. Other bioaerosol sampling has been carried out with different automated methods (e.g. Kawashima et al. 2016, Savage et al. 2017) and specifically in Europe with the Wibs system (e.g. O'Connor et al. 2015, Calvo et al. 2018).*

Thank you to reviewer 1 for pointing us towards these additional references. We have included further text to the introduction to describe the work done in these studies as well as the principles behind the various technologies used.

Automatic pollen monitoring is part of a broader field of research on automatic bioaerosol monitoring (Kawashima et al., 2017;Calvo et al., 2018; O'Connor et al., 2015; Savage et al., 2017), that was the object of a recent review article (Huffman et al., 2019).

*3.2. Pollen classification Lines184….. "Note that in this regard the problem presented to the algorithm is somewhat artificial: Corylus and grass pollen are not likely to be simultaneously present in the atmosphere in concentrations relevant for pollen allergies". This assertion is not entirely true, there are European regions where hazelnut and grass blooms coincide. On the other hand, in this work the Betula pollen type is not mentioned, with morphological characteristics and of dimensions practically equal to Corylus and that if it is present in the atmosphere at the same time as grasses, in very high concentrations and sufficient to cause allergic responses. It is not likely that Swispoleno can differentiate these types through digital holography. But it would be very important to point out this fact, since Betula is the pollen type that causes the most allergy in Central Europe, along with the grasses.*

We completely agree with the reviewer about this point and it will be essential that in future Betula is included the identification algorithm. This is likely to not be a straight-forward task but we are currently working on this aspect. We have added sentences to this section in this vein (see below).

In this line, it will be essential to include birch in the identification algorithm. This may, however, prove to be challenging given the morphological similarities of the members of the Betulaceae family.

*3.3 Reference particle counts and fluorescence observations Line 205-208. "At present, this remains a difficult task since no method, standardised or other, exists to aerosolise a known quantity of a known pollen taxa". "Pollen grains are both considerably larger than other, non-biological aerosol particles and relatively fragile, so producing homogenised airborne concentrations is currently not possible with conventional techniques". In relation to these two paragraphs and others of the same section 3.3, the authors should review in greater depth certain bibliography related to the methods of differentiation of biological particles and quantification (among other pollen and spores) by fluorescence (e.g. Toprak and Schnaiter 2013, Hernández et al. 2016, Savage et al. 2017).*

We thank the reviewer for pointing us towards these studies. The main aim of the sentences quoted was to point out that there is currently no standardised way to calibrate automatic bioaerosol detectors using fresh pollen particles (as, for example, is done for air quality monitoring devices). This has been clarified in the text and additional descriptions of the quantification and identification of biological particles added to the introduction.

*4 Towards operational pollen monitoring*
*"The focus of this study was to assess the performance of the Swisens Poleno, the first operational automatic pollen monitoring system based on digital holography". Certainly digital holography is the first time it has been used in bioparticle monitoring systems (specifically pollen) in real time, but the authors need to make a deep and detailed discussion, highlighting the advantages, similarities or great differences with other types of bioaerosol monitoring systems in real time. As I point out with some bibliographical references, certain pioneering works in this field are not mentioned. Finally, making few modifications to the introduction and discussion, the article can be published without problems, because I consider that the work constitutes a good scientific contribution, and a good basis to continue working in this field, as the authors propose.*

Again, we would like to thank reviewer 1 for highlighting the need for more detailed description of the literature. We agree that there has been extensive work done to identify biological particles using fluorescence and that these techniques may contribute to improve the identification algorithm of the Swisens Poleno. This a work in progress. We have added text to the conclusions in this regard:

Although the use of holography is a clear novelty of the present work, development of the method to additionally include florescence would build upon pioneering work performed using other devices (Toprak and Schnaiter, 2013; Hernandez et al., 2016; Savage et al., 2017).

**Reviewer 2**:
*This article presents the development and testing of a commercial instrument to characterize pollen aerosol particles with fluorescence and digital holography. The holography aspect is based on previous work by Berg & Videen in 2011 where images of similar free-flowing pollen particles are obtained revealing the particle size and shape. The authors integrates this powerful technique with fluorescence to collect material information from the particles at a later stage of trajectory through the instrument. This is important work as it demonstrates that the earlier proof-of-principle research can be effectively*

*implemented as an instrument. The paper's presentation is of high quality and its impact will be substantial to the field of aerosol science.*

Many thanks to reviewer 2 for the positive feedback, we really appreciate it!